# Tooth Formation as Experimental Model to Study Chemotherapy on Tissue Development: Effect of a Specific Dose of Temozolomide/Veliparib

**DOI:** 10.3390/genes13071198

**Published:** 2022-07-04

**Authors:** Sali Al-Ansari, Rozita Jalali, Antonius L. J. J. Bronckers, Olaf van Tellingen, Judith Raber-Durlacher, Nasser Nadjmi, Alan Henry Brook, Jan de Lange, Frederik R. Rozema

**Affiliations:** 1Department of Oral Medicine, Academic Center for Dentistry, 1081 LA Amsterdam, The Netherlands; sali.al-ansari@student.uantwerpen.be (S.A.-A.); j.raber.durlacher@acta.nl (J.R.-D.); 2Department of Cranio-Maxillofacial Surgery, Antwerp University Hospital, Antwerp, 2650 Edegem, Belgium; nasser.nadjmi@uza.be; 3Department of Oral Cell Biology, Academic Centre for Dentistry, 1081 LA Amsterdam, The Netherlands; rozita_jalali@yahoo.com (R.J.); bronckerst@gmail.com (A.L.J.J.B.); 4Department of Clinical Chemistry/Preclinical Pharmacology, Antoni van Leeuwenhoek-The Netherlands Cancer Institute, 1066 CX Amsterdam, The Netherlands; o.v.tellingen@nki.nl; 5Department of Oral and Maxillofacial Surgery, UMC, University of Amsterdam, 1081 LA Amsterdam, The Netherlands; j.delange@amc.uva.nl; 6School of Dentistry, University of Adelaide, Adelaide 5005, Australia; alan.brook@adelaide.edu.au; 7Institute of Dentistry, Queen Mary University of London, London E1 2AD, UK

**Keywords:** antineoplastic agents, dental epithelium, ion and fluid transport, gap junction, ameloblast, tooth formation

## Abstract

Background: Chemotherapy treatment of cancer in children can influence formation of normal tissues, leading to irreversible changes in their structure and function. Tooth formation is susceptible to several types of chemotherapy that induce irreversible changes in the structure of enamel, dentin and dental root morphology. These changes can make the teeth more prone to fracture or to caries when they have erupted. Recent studies report successful treatment of brain tumors with the alkylating drug temozolomide (TMZ) in combination with veliparib (VLP) in a glioblastoma in vivo mouse model. Whether these drugs also affect tooth formation is unknown. Aim: In this study the effect of TMZ/VLP on incisor formation was investigated in tissue sections of jaws from mice and compared with mice not treated with these drugs. Materials and method: The following aspects were studied using immunohistochemistry of specific protein markers including: (1) proliferation (by protein expression of proliferation marker Ki67) (2) a protein involved in *paracellular* ion transport (expression of tight junction (TJ) protein claudin-1) and (3) in *transcellular passage* of ions across the dental epithelium (expression of Na+, K+ 2Cl- cotransporter/NKCC1). Results: Chemotherapy with TMZ/VLP strongly reduced immunostaining for claudin-1 in distal parts of maturation ameloblasts. No gross changes were found in the treated mice, either in cell proliferation in the dental epithelium at the cervical loop or in the immunostaining pattern for NKCC1 in (non-ameloblastic) dental epithelium. The salivary glands in the treated mice contained strongly reduced immunostaining for NKCC1 in the basolateral membranes of acinar cells. Discussion/Conclusions: Based on the reduction of claudin-1 immunostaining in ameloblasts, TMZ/VLP may potentially influence forming enamel by changes in the structure of TJs structures in maturation ameloblasts, structures that are crucial for the selective passage of ions through the intercellular space between neighboring ameloblasts. The strongly reduced basolateral NKCC1 staining seen in fully-grown salivary glands of TMZ/VLP-treated mice suggests that TMZ/VLF could also influence ion transport in adult saliva by the salivary gland epithelium. This may cause treated children to be more susceptible to caries.

## 1. Introduction

With the application of newly developed anticancer agents, precision of dosage and early diagnosis, the life expectancy of patients with cancer, especially children, is prolonged. As more children survive, they may, in the long run, experience undesired negative side effects of this treatment on other tissues [1,2,3]. These changes could lead to irreversible changes in normal tissues, for instance defects in dental structures [2,4,5,6,7,8,9]. The extent and nature of oral complications vary in each patient and depend on the type of malignancy, type of cytostatic anticancer treatments used either alone or in combination with each other and to patient-related factors such as genetic factors, oral hygiene and oral health status [10]. As for chemotherapy, the toxicity of the antineoplastic agent depends on the nature, the therapeutic regimen, dose and duration of the treatment [11,12]. In children in whom tissues and organs are still developing, chemotherapy can affect formation of enamel and dentin by impairing activity of ameloblasts and odontoblasts, inhibit root formation or delay tooth eruption depending on the sensitivity of each of the involved cell types, the nature, dose and duration of the cytostatic [13,14,15]. In mature tissues, for example in functional salivary glands, chemotherapy can damage salivary epithelium, reducing production of saliva (hyposalivation) and change the composition, viscosity and buffer capacity of saliva. Such quantitative or qualitative changes in saliva could after tooth eruption, disrupt physicochemical balance between demineralization (white spot formation, a porosity due to local dissolution of crystals when pH in saliva drops by acidification by oral microorganisms) and remineralization of enamel subsurface layers by oral fluid. These white spots can either develop further into a caries lesion or disappear, having recovered by remineralization by mineral ions in saliva. The more acidic dental plaque found after chemotherapy could provide more favorable conditions for the emergence of a cariogenic flora [16,17,18].

The most common primary brain tumor is the glioblastoma (GBM). Treatment of such tumors is primarily by surgical resection with postsurgical therapy consisting of the DNA-alkylating agent temozolomide (TMZ) combined with radiotherapy. Even with such aggressive treatment, the prognosis of patients with GBM remains unsatisfactory. This is due to the function of PARP-1 and PARP-2 in base excision and DNA repair. Additional chemotherapy was necessary. PARP inhibitors (like veliparib) can sensitize GBM cells to TMZ and overcome the tumor’s resistance.

Temozolomide (TMZ) is an alkylating agent taken orally and used as first-line treatment of glioblastoma multiform as well as for recurrent anaplastic astrocytoma because of its ability to cross the blood–brain barrier (BBB) [19,20]. Because of its small size, TMZ easily crosses the blood–brain barrier after oral administration. TMZ [21] acts by the formation of nicks in the DNA structure during cell division which is followed by apoptosis [22]. This nonselective agent can affect both cancerous and normally-proliferating cells. However, since cancer cells divide more rapidly than normal cells, they should be more sensitive to these effects.

The most common side effect of TMZ is myelosuppression leading into thrombocytopenia and lymphopenia. These side effects are more likely with high doses. Any decrease in TMZ sensitivity is thus managed by combined therapy. The development of drug resistance is a major issue with TMZ treatment. Some of the known mechanisms of drug resistance may include intrinsic genetic or epigenetic factors as well as extrinsic factors.

O6 -methylguanine-DNA methyltransferase (MGMT) counteracts the action of TMZ. It repairs damaged DNA by eliminating alkyl groups produced by the alkylating agents such as TMZ [23]. The level of MGMT expression can be related to TMZ resistance [24,25]. Contradicting results were also reported [26]. However, miR-29c suppresses MGMT expression, and its overexpression increased TMZ efficacy. Consequently, miR-29c was suggested as a potential therapeutic target for glioma treatment [27].

Another mechanism of resistance is the mutation or inactivation of the DNA mismatch repair (MMR) system which is involved in the processing of DNA damage induced by TMZ [28].

In addition, the effect of TMZ is quickly and efficiently repaired by the base excision repair (BER) system. Within this system, poly (ADP-ribose) polymerase1 (PARP-1) is involved in recognition of the DNA damage [29] and consequently its repair. PARP inhibition improves TMZ’s in vitro and in vivo cytotoxicity [30].

GBM treatment resistance can also be explained by the presence of cancer cells with stem-like properties. The glioma stem cells (GSCs) are found among the tumor cell population. [31].

Acquired chemoresistance is a consequence of genetic and epigenetic changes induced by TMZ in neoplastic cells. GBM cells after TMZ therapy exhibited a gene expression program that differs between sensitive and resistant cells. This involves transcription factors, mRNAs, miRNAs, and lncRNAs [32,33]. In addition, mutation in telomere maintenance and telomerase activity was also implicated [34].

TMZ was tried in combination with other drugs to treat GBMs. In combination with trans sodium crocetinate (TSC), a drug that enhances oxygen delivery, a small beneficial effect was noted [35]. Some beneficial effects were reported by combining of TMZ with the addition of tumor-treating fields (TTFields) [36].

Other studies combining TMZ with bevacizumab (inhibitor of the Sonic hedgehog pathway) reported no significant benefits [37]. The addition of veliparib (a PARP inhibitor) to TMZ was shown by many studies to improve the efficacy of TMZ in treatment of GBM [38,39,40,41,42].

Veliparib (VLP), on the other hand, is an experimental anticancer agent, which is in clinical trial for treatment of various human malignancies including brain cancer (Clintrial.gov: NCT03581292). This drug inhibits poly ADP ribose polymerase (PARP), an enzyme involved in DNA repair. It is assumed that cancer cells will be more dependent on PARP than sound cells. VLP has shown promising results in experimental models of the brain [38,43].

To obtain a better understanding which sound tissues can be affected by cytostatics and could lead, in the long term, to dysfunction or structural defects requires testing of these agents on *normal* healthy tissues. To date, no information has been published as to whether enamel formation is sensitive to TMZ/VLP. In humans, formation of teeth occurs in a restricted period from 6 weeks in utero to 20 to 22 years. In small rodents, the incisor teeth are continuously produced life-long to replace loss of enamel and dentin at the incisal end due to abrasion. This makes it possible to study tooth development and effects of chemotherapy on incisor growth in adult mice.

In the developing rodent incisor cell, proliferation occurs in the cervical loop epithelium cells, followed by differentiation into secretory and maturation ameloblasts that transport many ion types to form apatite crystals and buffer the enamel fluid, enabling intramembranous transporters to transport NKCC1. Enamel formation and mineralization also requires the presence of tight junctions (TJ) (structures containing claudins and acting as barriers between neighboring ameloblasts to pass ions intercellularly). Transcellular transport of Na^+^, Cl^−^ and K^+^ by the non-ameloblast dental epithelium is regulated by Na^+^, K^+^ and 2Cl cotransporter 1 (NKCC1) and is essential for full completion of mineralization of enamel [44].

The aim of the present study is to examine the effect of TMZ/VLP on ameloblasts in continuously developing mouse incisors.

## 2. Materials and Methods

### 2.1. Experimental Animals and Tissues

For this analysis, we used tissue material collected from young adult mice 8-15 weeks old that were used in efficacy studies with TMZ and/or VLP against experimental glioblastoma [38]. Veliparib (ABT-888) was obtained from Selleck Chemicals and TMZ from TEVA Pharma. For the present study, (archival) blocks of paraffin- embedded heads of the same mice were used, all inbred Abcg2; Abcb1a/b knockout mice grafted with GBM652457 tumor cells injected into the brain. The experimental mice had been treated for 5 days with a mixture of TMZ 100 mg/kg/QD and VLP 10 mg/kg/BID, whereas vehicle-treated animals served as controls (summary in Table 1). Drug plasma and brain samples were analyzed by liquid chromatography/tandem mass spectrometry (LC/MS-MS) as described in detail in previous publications (see Lin et al., 2014 [38]).

The doses used are correlated to clinically relevant exposure. The aim of this study was to investigate the effects of normal-range therapeutic doses of these drugs on teeth development. Further studies may be needed to explore the effects of doses of very low or high toxicity.

Twelve to seventeen days after the last injection the animals were euthanized. Animals were humanely killed when reaching the endpoint [23]. Following the removal of the skin, the complete heads were immersed in ethanol; acetic acid; formalin (EAF) fixative. After decalcification in formic acid slices of about 3 mm were embedded in paraffin.

All experiments involving animals were approved by the local animal ethics committee (see [23]).

### 2.2. Processing of Tissues and Histology

All tissue blocks were processed into 5–7 µm-thick paraffin sections mounted on glass slides. Dewaxed sections were stained with 1% hematoxylin (1 min) and eosin (5 min) (HE) or used for immunohistochemical staining.

### 2.3. Immunohistochemistry

The paraffin sections were dewaxed in xylene, rehydrated in a descending series of ethanol concentrations, and rinsed in phosphate-buffered saline (PBS). Sections were subjected to antigen retrieval in 10mM citrate buffer (pH 6.0) either at 60 °C overnight or for 20 min in a microwave at 95 °C. Endogenous peroxidase was blocked with a peroxidase block solution (Envision kit, Dakocytomation, Glostrup, Denmark) for 5 min. Sections were washed three times in tris-buffered saline (TBS). Non-specific staining was blocked for 30 min with 2% BSA after which sections were incubated overnight at 4 °C with primary antibodies. These were (1) goat anti-NKCC1 (Santa Cruz, N-16, affinity purified, catalog number SC-21545), raised against the N-terminal end of human NKCC1. (2) rabbit anti-claudin-1 (Abcam, ab #15098) and (3) rabbit anti-Ki67 (Abcam, ab#15580; dilution 1:200–1:300) The Ki67 nuclear antigen is expressed in the cell cycle phases G1, S, G2 and M, but is absent in G0. It localizes to the perinucleolar region during G1. In (4), matched non-immune IgG (1:200–1:300) or normal serum (same concentration as primary antibodies) served as controls. After overnight incubation at 4 °C with primary antibodies, sections were washed three times in TBS and incubated with rabbit anti-goat secondary antibody conjugated to peroxidase (Thermo Scientific, 168 Third Avenue, Waltham, MA, USA 02451) or goat anti-rabbit (Envision kit) for 1 h at room temperature. After washing, staining was visualized using Diaminobenzidine (DAB; Envision kit) and counterstained with hematoxylin. Immunohistochemistry images were acquired with a Leica EL6000 or Axio Zoom V16 microscope. The evaluation of the sections was performed double blind to avoid any bias.

## 3. Results

Histological evaluation of sections of lower mouse jaws stained with hematoxylin- eosin (H&E) showed no difference in cell and tissue structure of secretory ameloblasts or odontoblasts between the control (Figure 1a) and experimental (Figure 1b) group. The layers of the inner dental epithelium in the cervical loop, and its differentiation into secretory ameloblasts, looked normal in both experimental and control groups. In the cervical loop of both groups dental epithelium cells were positive for cell proliferation marker Ki67 without clear differences (Figure 2a,b). However, in the experimental group, the position of the nuclei in some maturation stage ameloblasts was more central than basal (Figure 1d). No major changes were noted in the structure of dentine, odontoblasts or pulp after chemotherapy treatment (Figure 1a–d).

An intense signal for claudin-1 (a marker for tight junctions) was seen in the control group as a discrete brown-stained line in the distal parts (arrows in Figure 2c,e) and weaker intracellularly in the supranuclear part of maturation ameloblasts in the control group (Figure 2c,e). No such positive staining for claudin-1 in the distal part of maturation ameloblasts was noticed in mice that received chemotherapy (Figure 2d,f) but strong intracellular staining in groups of maturation ameloblasts (Figure 2d,f) was evident. Also, weak claudin-1 staining was noted in dental epithelium that was in contact with the maturation ameloblasts (Figure 2f).

In developing teeth of both experimental and control mice the membranes of the cells of the epithelial papillary layer (overlaying the ameloblast layer), but not ameloblasts, immunostained strongly for Na^+^:K^+^:2Cl^−^ cotransporter (NKCC1) without a clear difference in distribution or intensity of staining. Interestingly, in salivary glands of the mice that received chemotherapy, NKCC1 staining in the basolateral membranes of the acinar cells was strongly reduced (Figure 3c,d).

## 4. Discussion

The aim of chemotherapy is to inhibit the proliferation of cancer cells by blocking DNA and/or protein synthesis in cancer cells. However, often the drugs lack specificity, and influence also the functioning of sound cells/tissues. When applied during childhood, chemotherapy can disrupt normal tissue development. In developing teeth this can result in irreversible changes in the structure and function of enamel or dentin.

Our data suggest that under the given conditions (dose, application and duration), which relate to those used in the treatment of children, TMZ/VLP has no gross effects on cell proliferation, differentiation into secretory ameloblasts or on NKCC1-mediated ion-transport by the overlaying (non-ameloblast) dental epithelium. In contrast, the protein expression of claudin-1 was seen in the distal parts of maturation ameloblasts in controls but was absent in the TMZ/VLP-treated mice. The major function of claudins is enabling selective passage of ions into, or from, the luminal space of transport epithelia (review Gong and Hou 2017) [45].

Claudin-1 is one of a family of 24 claudin isotypes specific for TJ in many ion-transport epithelia that act as physical barriers for most ion types but selectively enable some ion types to pass. In renal thick ascending loop of Henle complexes of claudin 10b or complexes of claudin-16 and claudin-19 form channels in tight junctions that differ in size, 3D structure and charge, and enable selective passage of ions through the intercellular space [45,46,47]. Claudin-10b, for example, forms Na+ channels and claudin-16 and -19 complexes form cation channels [46,47]. Ameloblasts express several claudin isoforms, most interestingly claudin-1, -3, -16 and -19 [48,49,50,51,52]. Absence of claudin-3, -16 and -19 in mouse null mutants has been associated with development of enamel defects [50,51,52]. Whether claudin-1 is also essential for amelogenesis is unknown. The absence of claudin-1 in distal membranes of maturation ameloblasts in TMZ/VLP-treated mice suggests an impaired or incomplete formation of tight junctions. These changes in claudin-1 could be associated with mineralization defects similar those reported for claudin-3, -16 and -19.

Uncontrolled passage of ions across the ameloblastic layer into forming enamel due to defects of TJ in ameloblasts can well influence pH and the levels of Na^+^, Cl^−^, Ca^2+^ and K^+^ in enamel fluid, which in turn delays or impairs enamel mineralization.

NKCC1 is an ion transporter for Na^+^, K^+^ and Cl^−^ involved in water transport. It is not expressed in ameloblasts, but in other epithelial cells of the enamel organ. NKCC1 is also localized in the basolateral membranes of parotid acinar cells, but not in duct cells. Lack of functional NKCC1 (*Nkcc1* null mice) results in a dramatic reduction (>60%) in the volume of saliva secreted in response to a muscarinic stimulus. In salivary glands, NKCC1 is required to transport ions and water enabling salivary flow [53]. In the present study, the NKCC1 staining in salivary cells was far less intense after chemotherapy in comparison to that in untreated control mice. It suggests that chemotherapy could reduce secretion of saliva, one of the reasons that patients often suffer from dry mouth after chemotherapy.

In NKCC1-deficient mice, enamel mineralization is reduced [53]. Why NKCC1 staining in dental (non-ameloblast) epithelium seemed unaffected by chemotherapy in contrast to salivary glands is unclear. The drugs may have reduced the production of NKCC1 protein in dental epithelium only slightly, an effect too small to be noticed by immunohistochemical staining.

For treatment of brain tumors, it is pertinent to mention that NKCC1 is also involved in transport across the blood–brain barrier, and that mutation of NKCC1 can result in dysfunction of the brain [54,55].

### 4.1. Mice as a Study Model

The incisors of small rodents produce enamel and dentin throughout their lifetime and produce a complete incisor in a relatively short time (5–6 weeks in mice). The complete life cycle of dental epithelium from proliferation in the cervical loop until differentiation into ameloblasts, formation of enamel, apoptosis and eruption of the mature enamel can be followed in a single histological sagitally-cut section. Disruptions of enamel formation can result in mineralization defects accumulating as lasting recordings. This makes the continuously erupting incisor of mice and rats an attractive model for testing cytostatic drugs on tissue development.

### 4.2. Chemotherapeutic Agents and Dental Development

Dental development is a complex adaptive system involving genetic, epigenetic and environmental factors [56]. These factors interact in networks over the long period of the progressive development of each tooth through the stages of initiation, morphogenesis, differentiation, and calcification [57]. The phenotypic outcome of disturbances in this process varies from congenitally missing or supernumerary teeth to very small or large teeth, to abnormally shaped teeth, and to defects of the mineralized tissues [58]. The findings of this study suggest that the environmental chemotherapeutic agents investigated here would probably interact in this complex developmental system to produce enamel mineralization defects when used in children and teenagers, since human dental development continues until 20 to 22 years of age.

### 4.3. Limitations of This Study

We assumed that the tissue in the control mice group (which also contains the tumor) was functioning normally and that the parotid gland and the incisor tissue were normal. The number of mice treated in this study was small, but even in this limited sample the abnormality was clearly visible. This study used materials obtained in a previous study. The enamel defects in Claudin-3-, -16-, -19-null mice and the presence of claudin-1 in ameloblast cells suggests that absence of ng claudin-1 in tight junctions is a possible cause of abnormalities in the enamel development. This possibility could not be excluded as additional control tissue of mice not injected with tumor cells control tissue was not available

## 5. Conclusions

We conclude that the use of chemotherapeutic agents (TMZ/Veliparib) in the treatment of brain tumors in children can affect the development of teeth and oral structures, even when they are given in the usual and recommended dosage. Children under such a regimen of cancer treatment should be followed up carefully and their teeth and oral tissues regularly examined, so that any necessary dental treatment can be provided at the optimum time, and side effects to be dealt with correspondingly.

## Figures and Tables

**Figure 1 genes-13-01198-f001:**
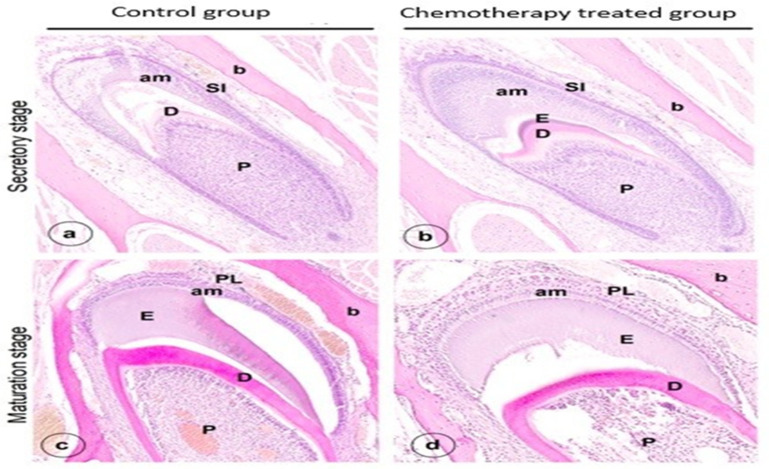
H&E-stained section of developing mouse incisors. Cross sections containing secretory stage (**a**) and maturation stage ameloblasts (**c**) of a control mouse. Sections of secretory stage (**b**) and maturation stage ameloblasts (**d**) of an experimental mouse: PL, papillary layer; P, pulp; am, ameloblasts; D, dentin; E, enamel; b, bone; SI, stratum intermedium, Cl cervical loop.

**Figure 2 genes-13-01198-f002:**
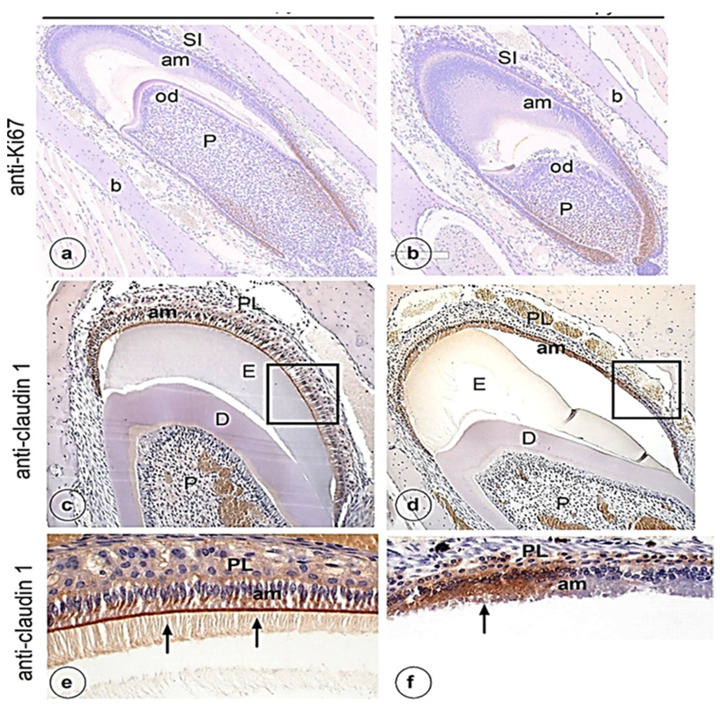
Ki-67 (**a**,**b**) and Claudin-1 (**c**–**f**) immunostaining of ameloblasts from mandibular incisor in control (**a**,**c**,**e**) and experimental mice (**b**,**d**,**f**): PL, papillary layer; P, pulp; am, ameloblasts; D, dentin; E, enamel; b, bone; SI, stratum intermedium. Cl cervical loop.

**Figure 3 genes-13-01198-f003:**
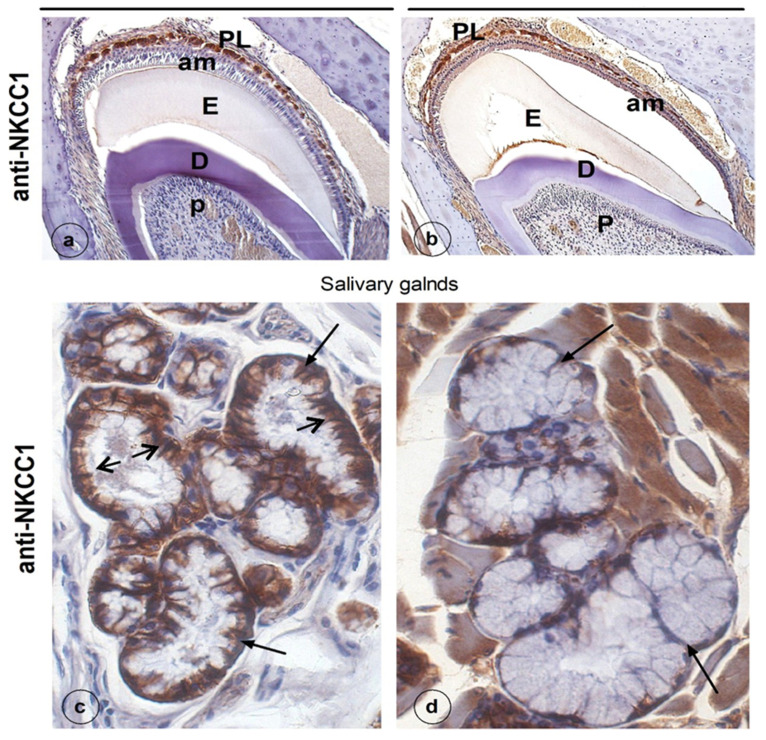
NKCC1 immunostaining of cells of the papillary layer but not in ameloblasts (**a**,**b**). Basolateral staining for NKCC1 in salivary glands (**c**,**d**) was found in the control (**a**,**c**) but not after chemotherapy (**d**): PL, papillary layer; P, pulp; am, ameloblasts; D, dentin; E, enamel; b, bone; SI, stratum intermedium.

**Table 1 genes-13-01198-t001:** Summary of the study set-up, animals, procedures and materials.

Group	Tumor	Number Mice and Sex	Age of Mice at Start (Weeks)	Dose per Day (iv) (One Injection/Day)	Duration of Treatment (Days)
Experimental	Yes	3 female	8–15 weeks	100 mg/kg TMZ + 25 mg/kg VLP	5 days
Control	Yes	2 female	8–15 weeks	Vehicle only	5 days

## Data Availability

Data can be found (2 locations): 1. Department of Oral Medicine, Academic Center for Dentistry, 1081 LA Amsterdam, The Netherlands; 2. Department of Clinical Chemistry/Preclinical Pharmacology, Antoni van Leeuwenhoek-The Netherlands Cancer Institute, 1066 CX Amsterdam, The Netherlands.

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
