# Peer review of "Tooth Formation as Experimental Model to Study Chemotherapy on Tissue Development: Effect of a Specific Dose of Temozolomide/Veliparib"

_genes, 2022, doi:10.3390/genes13071198_

Round 1

Reviewer 1 Report

In this study, the authors selected two anticancer drugs temozolomide/veliparib to investigate the potential effect on tissue (tooth) development. The authors chose mice as the model organism (Abcg2;Abcb1a/b knockout mice grafted with GBM652457 tumor cells into brain) and choose a specific dose for each drug for injection. They performed immunostaining and showed no major changes of immunostaining pattern for NKCC1 or cervical loop but strongly reduced staining for NKCC1 in basolateral membranes of acinar cells of salivary glands. While this study seems interesting and the results provide some insights into the tissue formation after temozolomide/veliparib treatment, the study is not carefully designed and the results may need to be confirmed by additional experiments. My specific comments are below.

The rationale of studying temozolomide/veliparib on tooth formation and cell proliferation is not clearly described in the Abstract and main text. Temozolomide is used for treatment of glioblastoma and other brain tumors.  Why is studied here for tooth formation? And why studied the combination of these two drugs, not other drug combination?

Only one specific dose for each drug is applied in this study (Table 1). It typically needs a serious of doses to find what scenario of drug treatment may most affect the phenotype and tissue development.

The mice at age 8-15 weeks are not young. Why chose such adult mice, not younger ones?

In Introduction, lines 74-81, the authors may add more introduction of temozolomide for treatment of GBM and how likely this affects the outcome. The authors may consider adding more background by referring literature like these publications (Fritah et al. 2020, PMID: 32927769; Liu et al. 2016, PMID: 27784795).

Author Response

We thank the reviewers for their efforts and recommendations to improve our manuscript. We believe that all the remarks raised by the reviewers were valid. We discuss below point by point our answers actions taken to meet their suggestions:

Reviewer 1

In this study, the authors selected two anticancer drugs temozolomide/veliparib to investigate the potential effect on tissue (tooth) development. The authors chose mice as the model organism (Abcg2;Abcb1a/b knockout mice grafted with GBM652457 tumor cells into brain) and choose a specific dose for each drug for injection. They performed immunostaining and showed no major changes of immunostaining pattern for NKCC1 or cervical loop but strongly reduced staining for NKCC1 in basolateral membranes of acinar cells of salivary glands. While this study seems interesting and the results provide some insights into the tissue formation after temozolomide/veliparib treatment, the study is not carefully designed and the results may need to be confirmed by additional experiments. My specific comments are below.

Reviewer 1, Question1: The rationale of studying temozolomide/veliparib on tooth formation and cell proliferation is not clearly described in the Abstract and main text. Temozolomide is used for treatment of glioblastoma and other brain tumors.  Why is studied here for tooth formation? And why studied the combination of these two drugs, not other drug combination?

Answer by the authors:

We added to the abstract why this study was carried out and also made this clearer in the introduction.  This study was a follow up of our previous report that examined the effect of both drugs on a brain tumor model. The setup, treatment and dose were explained in that paper. We used the tissue blocks collected in the previous report to examine incisor development.

In Discussion section more background was given and special focus was offered to claudin-1; new references (8 new references) to tight junctions and claudin -1 were inserted to discuss the changes in claudin-1 in ameloblasts induced by TMZ/VLP.

The Introduction section was revised and rewritten according to the suggestions of the reviewers

Why the combination of TMZ/VLP?

Brain tumors are common in pediatric patients. They are primarily treated by the alkylating agent temozolomide which acts by inducing DNA damages. Some patients showed poor results and relapses (Pietanza et al. 2012). This was attributed to DNA repair by PARP and represents a resistance mechanisms to temozolomide. Based on this hypothesis the efficacy of temozolomide is increased when combined with the PARP inhibitor veliparib (Lazzari et al. 2018)

References:

- Pietanza MC, Kadota K, Huberman K, et al. Phase II trial of temozolomide in patients with relapsed sensitive or refractory small cell lung cancer, with assessment of methylguanine-DNA methyltransferase as a potential biomarker. Clin Cancer Res 2012;18:1138-45.

- Chiara Lazzari, Vanesa Gregorc, Alessandra Bulotta, Alessia Dottore, Giuseppe Altavilla, and Mariacarmela Santarpia

Temozolomide in combination with either veliparib or placebo in patients with relapsed-sensitive or refractory small-cell lung cancerTransl Lung Cancer Res. 2018 Dec; 7(Suppl 4): S329–S333.

Reviewer 1, Question 2 Only one specific dose for each drug is applied in this study (Table 1). It typically needs a serious of doses to find what scenario of drug treatment may most affect the phenotype and tissue development. The mice at age 8-15 weeks are not young. Why chose such adult mice, not younger ones?

Answer by the authors 

The drugs used and the dosage:

Temozolomide (TMZ) is used for brain tumors which are the second most common cancer in children after leukemia. Therefore, those children receive temozolomide at a young age and we aimed to investigate the effect of this drug on tooth formation which is occurring in this age group. To our knowledge. there is no research done on the effect of temozolomide/Veliparib on tooth formation.

This dose in this experiment was used in the previous report with mice, per kg weight and was based on the dose used to treat brain tumors.

In this study, the aim was to investigate the effects of normally used anticancer drugs’ doses (i.e not other doses) on the teeth and oral tissues growth and development.

The animals’ age:

Mice 8-15 weeks old are young adult with most tissues fully developed (e.g. salivary glands) and developing tissues (e.g. renewing intestinal lining and incisor teeth that continuously erupting throughout life. The incisor teeth of mice grow continually throughout life and are therefore a suitable model for examining effects during tooth development: in humans tooth development occurs from 6 weeks in utero until 20 to 22 years of age. In the mouse the turnover of the entire tooth in 35-45 days (Garcia-Aorocina 2016).

References:

Diagnosis | Severe prognathic malocclusion. Lab Animal volume 36, pages22–23 (2007) https://doi.org/10.1038/laban0107-22

Dolores Garcia-Arocena, Ph.D. How to spot and manage malocclusion in research mice. The Jackson Laboratory, Blog Post July 20; 2016.

Reviewer 1, Question3: In Introduction, lines 74-81, the authors may add more introduction of temozolomide for treatment of GBM and how likely this affects the outcome. The authors may consider adding more background by referring literature like these publications (Fritah et al. 2020, PMID: 32927769; Liu et al. 2016, PMID: 27784795).

This point was addressed, and the following paragraph was added in Introduction after the first paragraph:

The most common primary brain tumor is the Glioblastoma (GBM). Treatment of such tumors is primary by surgical resection with postsurgical therapy consisted of the DNA-alkylating agent temozolomide (TMZ)-combined with radiotherapy. Even with such aggressive treatment the prognosis of patients with GBM remains unsatisfactory. This is due to the function of PARP-1 and PARP-2 in base excision and DNA repair. Therefore, additional chemotherapy is essential. PARP inhibitors (like veliparib) can sensitize GBM cells to TMZ and overcomes the tumors resistance. 

Reviewer 2 Report

The authors aimed to evaluate the effects of the mixture of temozolomide (TMZ), in combination with veliparib (VLP) on cells that form incisor teeth in mice. In humans, formation of teeth occurs only in children and teenagers. The data suggest that in developing mouse incisors TMZ in combination with VLP may influence the barrier function of maturation ameloblasts by changing the composition of tight junctions which may result in enamel defects.

The study covers some issues that have been overlooked in other similar topics. The structure of the manuscript appears adequate and well divided in the sections. Moreover, the study is easy to follow, but some issues should be improved. Some of the comments that would improve the overall quality of the study are:

a. Authors must pay attention to the technical terms acronyms they used in the text.

b. English language needs to be revised.

c. Limitations of the study needs to be added.                      

d. Conclusion Section: This paragraph needs to be added, including some "take-home message".

Author Response

We thank the reviewers for their efforts and recommendations to improve our manuscript. We believe that all the remarks raised by the reviewers were valid. We discuss below point by point our answers actions taken to meet their suggestions:

Reviewer 2

The authors aimed to evaluate the effects of the mixture of temozolomide (TMZ), in combination with veliparib (VLP) on cells that form incisor teeth in mice. In humans, formation of teeth occurs only in children and teenagers. The data suggest that in developing mouse incisors TMZ in combination with VLP may influence the barrier function of maturation ameloblasts by changing the composition of tight junctions which may result in enamel defects.

The study covers some issues that have been overlooked in other similar topics. The structure of the manuscript appears adequate and well divided in the sections. Moreover, the study is easy to follow, but some issues should be improved. Some of the comments that would improve the overall quality of the study are:

  1. Authors must pay attention to the technical terms acronyms they used in the text. 

Action: Done

  1. English language needs to be revised

Action: Done

  1. Limitations of the study needs to be added. 

Limitations: Done, and the following paragraph is added to the discussion section

Limitations of this study:

We assumed that the tissue in the control mice group (which also contains the tumor) was functioning normally and that the parotid gland and the incisor tissue was normal. The number of mice treated in this study is small, but even in this limited sample the  abnormality was clearly visible.

This study used material obtained in a previous study. The enamel defects in Claudin-3, 16, 19 null mice and the presence of claudin-1 in ameloblast cells suggests that absence of ng claudin-1 in tight junctions is a possible cause of abnormalities in the enamel development. This possibility could not be excluded as control tissue was not available.  

  1. Conclusion Section: This paragraph needs to be added, including some "take-home message".

Done, and the following paragraph was added at the end of the discussion

We conclude that the use of chemotherapeutic agents (TMZ/ Veliparib) in the treatment of brain tumors in children can affect the development of teeth and oral structures, even when they are given in the usual and recommended dosage. Children under such a regimen of cancer treatment should be followed up carefully and their teeth and oral tissues regularly examined, so that any necessary dental treatment can be provided at the optimum time.

Round 2

Reviewer 1 Report

The authors have addressed some of my concerns, mostly by explanation and added some background text in the revised manuscript. Specifically, the explanation of the rationale is much clearer now in the revised manuscript.

1.     Regarding the combination of two drugs, the authors had good explanation. It is better to explain why these two specific drugs as the combination. This should be an easy explanation by the authors if the study is carefully thought.

2.     Regarding the previous comment “the authors may add more introduction of temozolomide for treatment of GBM and how likely this affects the outcome. The authors may consider adding more background by referring literature like these publications (Fritah et al. 2020, PMID: 32927769; Liu et al. 2016, PMID: 27784795).” The authors added some general introduction, without citing any references (e.g. Fritah et al. 2020, PMID: 32927769; Liu et al. 2016, PMID: 27784795). Please cite references. It also could be broader (rather than PARP-1 and PARP-2) since TMZ has been well studied in GBM.

3.     Regarding dose scheme in the study design, the authors only explained their specific dose that was supported from the previous report (btw, what report?, please specify and cite the reference). However, different doses are always required to find the related and appropriate drug response outcome. This is standard in the field. The authors did not respond this comment satisfactory. Their results can be potentially misleading if the dose(s) are not representative or appropriately administrated.

Author Response

We appreciate  the positive comments  concerning the improvements made to  our paper following Reviewer 2’s initial report. We also appreciate that the points stressed in the second review will  make our manuscript more solid and of higher value. We agree on these points and have further revised  our paper  by editing some paragraphs as suggested.

Here we answer  point by point, the  issues raised by the reviewer and indicate the changes made in the manuscript. We hope these changes will make our article acceptable for publication in Genes:

Points 1 and 2: More details about TMZ, its sensitivity, combination therapy with veliparib:

Answer:

We realized that more information is needed. A new paragraph is added to the introduction dealing with all aspects raised by the reviewer.  Twenty new references are  added including the two references suggested by the reviewer:

The most common side effect of TMZ is myelosuppression leading into thrombocytopenia and lymphopenia. These side effects are more likely with high doses. Any decrease in TMZ sensitivity is thus managed by combined therapy. The development of drug resistance is a major issue with TMZ treatment. Some of the known mechanisms of drug resistance may include intrinsic genetic or epigenetic factors as well as extrinsic factors.

O6 -methylguanine-DNA methyltransferase (MGMT) counteracts the action of TMZ. It repairs damaged DNA by eliminating alkyl groups produced by the alkylating agents such as TMZ [23]. The level of MGMT expression  can be related to TMZ resistance (24, 25).  Contradicting results were also reported [26]. However, miR-29c suppresses MGMT expression and its overexpression increased TMZ efficacy.  Consequently, miR-29c was suggested as a potential therapeutic target for glioma treatment [27].

Another mechanism of resistance is the mutation or inactivation of the DNA mismatch repair (MMR) system which is involved in the processing of DNA damage induced by TMZ [28].

In addition, the effect of TMZ is quickly and efficiently repaired by the Base Excision Repair (BER) system. Within this system, poly (ADP-ribose) polymerase1 (PARP-1) is involved in recognition of the DNA damage [29] and consequently its repair. PARP inhibition improves TMZ’s in vitro and in vivo cytotoxicity [30].

GBM treatment resistance can also be explained by the presence of cancer cells with stem like properties. The glioma stem cells (GSCs) are found among the tumor cell population. [31].

Acquired chemoresistance is a consequence of genetic and epigenetic changes induced by TMZ in neoplastic cells. GBM cells after TMZ therapy exhibited a gene expression program that differs between sensitive and resistant cells. This involves transcription factors, mRNAs, miRNAs, and lncRNAs [32, 33]. In addition, mutation in telomere maintenance and telomerase activity was also implicated [34].

TMZ was tried in combination with other drugs to treat GBMs. In combination with trans sodium crocetinate (TSC), a drug that enhances oxygen delivery, a small beneficiary effect was noted [35]. Some beneficial effects were reported by combining of TMZ with the addition of tumor-treating fields (TTFields) [36].

Other studies by combining TMZ with bevacizumab (inhibitor of the Sonic hedgehog pathway) reported no significant benefits [37]. The addition of veliparib (a PARP inhibitor) to TMZ was shown by many studies to improve the efficacy of TMZ in treatment of GBM [38- 42].

Point 3:

Answer: Actually, this research was done following  research performed by colleagues and published in clinical cancer research (Lin et al 2014) and cited here in our manuscript. That  paper discussed the effectiveness of combination of TMZ and veliparib on the treatment of GBM. We used the heads of these animals to study the possible effects of this combination therapy on teeth development. One therapeutic clinically advised dose was used. However, we have added  more explanation about the drugs and doses used. We emphasized that further future studies may be needed to explore the effects of more extreme doses. We also modified the title of this manuscript to express this point. The following paragraph is added to methodology:

Velparib (ABT-888) was obtained from Selleck Chemicals and TMZ from TEVA Pharma.

Drug plasma and brain samples were analyzed by liquid chromatography/tandem mass spectrometry (LC/MS-MS) as described in detail in previous publication (see Lin et al 2014)

The doses used are correlated to clinically relevant exposure. The aim of this study was to investigate the effects of normal range therapeutic doses of these drugs on teeth development. Further studies may be needed to explore the effects of very low or high toxic doses.

We changed the title into:

Tooth formation as experimental model to study chemotherapy on tissue development: Effect of a specific dose of temozolomide/veliparib

New References:

Marchesi, F.; Turriziani, M.; Tortorelli, G.; Avvisati, G.; Torino, F.; De Vecchis, L. Triazene compounds: mechanism of action and related DNA repair systems. Pharmacol. Res., 2007, 56(4), 275- 287

Thon, N.; Kreth, S.; Kreth, F.W. Personalized treatment strategies in glioblastoma: MGMT promoter methylation status. OncoTargets Ther., 2013, 6, 1363-1372. http://dx.doi.org/10.2147/OTT.S50208 PMID: 24109190

Parker, N.R.; Khong, P.; Parkinson, J.F.; Howell, V.M.; Wheeler, H.R. Molecular heterogeneity in glioblastoma: potential clinical implications. Front. Oncol., 2015, 5, 55. http://dx.doi.org/10.3389/fonc.2015.00055 PMID: 25785247

Felsberg, J.; Thon, N.; Eigenbrod, S.; Hentschel, B.; Sabel, M.C.; Westphal, M.; Schackert, G.; Kreth, F.W.; Pietsch, T.; Löffler, M.; Weller, M.; Reifenberger, G.; Tonn, J.C. German Glioma Network. Promoter methylation and expression of MGMT and the DNA mismatch repair genes MLH1, MSH2, MSH6 and PMS2 in paired primary and recurrent glioblastomas. Int. J. Cancer, 2011, 129(3), 659-670.

 Xiao,S. Yang.Z, Qiu. X, Lv. R, Liu.J, Wu.M, Liao.Y, Liu.Q. miR-29c contribute to glioma cells temozolomide sensitivity by targeting O6-methylguanine-DNA methyltransferases indirectly. 2016; Oncotarget, Vol. 7, No. 31 :50229-50237.

Marton, E.; Giordan, E.; Siddi, F.; Curzi, C.; Canova, G.; Scarpa, B.; Guerriero, A.; Rossi, S.; D’ Avella, D.; Longatti, P.; Feletti, A. Over ten years overall survival in glioblastoma: A different disease? J. Neurol. Sci., 2020, 408, 116518

Dantzer, F.; Amé, J.C.; Schreiber, V.; Nakamura, J.; Ménissier-de Murcia, J.; de Murcia, G. Poly(ADP-ribose) polymerase-1 activation during DNA damage and repair. Methods Enzymol., 2006, 409, 493-510. http://dx.doi.org/10.1016/S0076-6879(05)09029-4 PMID: 16793420

Kinsella, T.J. Coordination of DNA mismatch repair and base excision repair processing of chemotherapy and radiation damage for targeting resistant cancers. Clin. Cancer Res., 2009, 15(6), 1853-1859. http://dx.doi.org/10.1158/1078-0432.CCR-08-1307 PMID: 19240165

Abou-Antoun, T.J.; Hale, J.S.; Lathia, J.D.; Dombrowski, S.M. Brain Cancer Stem Cells in Adults and Children: Cell Biology and Therapeutic Implications. Neurotherapeutics, 2017, 14(2), 372-384. http://dx.doi.org/10.1007/s13311-017-0524-0 PMID: 28374184

Fritah. S, Muller.A, Jiang.W, Mitra.R, Sarmini.M, Dieterle.M, Golebiewska.A, Ye.T, Van Dyck.E, Herold-Mende.C, Zhao.Z, Azuaje.F, and Niclou.SP. Temozolomide-Induced RNA Interactome Uncovers Novel LncRNA Regulatory Loops in Glioblastoma.  Cancers 2020, 12, 2583; doi:10.3390/cancers12092583.

Liu.S, Mitra.R, Zhao.M-M, Fan.W, Eischen.CM, Yin.F, Zhao.Z. The Potential Roles of Long Noncoding RNAs (lncRNA) in Glioblastoma Development.Mol Cancer Ther. 2016;15(12):2977-2986. doi: 10.1158/1535-7163.MCT-16-0320. Epub 2016 Oct 26.

Cai.H-Q, Liu. A-S, Zhang.M-J, Liu. H-J, Meng. X-L, Qian.H-P and Wan.J-H. Identifying Predictive Gene Expression and Signature Related to Temozolomide Sensitivity of Glioblastomas. Frontiers in Oncology, 2020; Vol 10: 669.

Gainer, J.L.; Sheehan, J.P.; Larner, J.M.; Jones, D.R. Trans sodium crocetinate with temozolomide and radiation therapy for glioblastoma multiforme. J. Neurosurg., 2017, 126(2), 460-466. http://dx.doi.org/10.3171/2016.3.JNS152693 PMID: 27177177

Stupp, R.; Taillibert, S.; Kanner, A.A.; Kesari, S.; Steinberg, D.M.; Toms, S.A.; Taylor, L.P.; Lieberman, F.; Silvani, A.; Fink, K.L.; Barnett, G.H.; Zhu, J.J.; Henson, J.W.; Engelhard, H.H.; Chen, T.C.; Tran, D.D.; Sroubek, J.; Tran, N.D.; Hottinger, A.F.; Landolfi, J.; Desai, R.; Caroli, M.; Kew, Y.; Honnorat, J.; Idbaih, A.; Kirson, E.D.; Weinberg, U.; Palti, Y.; Hegi, M.E.; Ram, Z. Maintenance therapy with tumor-treating fields plus temozolomide vs temozolomide alone for glioblastoma: A Randomized Clinical Trial. JAMA, 2015, 314(23), 2535-2543

Gilbert, M.R.; Dignam, J.J.; Armstrong, T.S.; Wefel, J.S.; Blumenthal, D.T.; Vogelbaum, M.A.; Colman, H.; Chakravarti, A.; Pugh, S.; Won, M.; Jeraj, R.; Brown, P.D.; Jaeckle, K.A.; Schiff, D.; Stieber, V.W.; Brachman, D.G.; Werner-Wasik, M.; TremontLukats, I.W.; Sulman, E.P.; Aldape, K.D.; Curran, W.J., Jr; Mehta, M.P. A randomized trial of bevacizumab for newly diagnosed glioblastoma. N. Engl. J. Med., 2014, 370(8), 699-708.

Lin, F.; De Gooijer, M.C.; Moreno Roig, E.; Buil, L.C.M.; Christner, S.M.; Beumer, J.H.; Wurdinger, T.;

Beijnen, J.H.; Van Tellingen, O. ABCB1, ABCG2, and PTEN Determine the Response of Glioblastoma to

Temozolomide and ABT-888 Therapy. Clin Cancer Res. 2014, 20,2703-2713.   (ALREADY CITED BEFORE)

Pietanza, M.C.; Waqar, S.N.; Krug, L.M.; Dowlati, A.; Hann, C.L.; Chiappori, A.; Owonikoko, T.K.; Woo,

K.M.; Cardnell, R.J.; Fujimoto, J.; Long, L.; Diao, L.; Wang, J.; Bensman, Y.; Hurtado, B.; de Groot, P.;

Sulman, E.P.; Wistuba, I.I.; Chen, A.; Fleisher, M.; Heymach, J.V.; Kris, M.G.; Rudin, C.M.; Byers, L.A. Randomized, double-blind, phase ii study of temozolomide in combination with either veliparib or placebo in patients with relapsed-sensitive or refractory small-cell lung cancer. J. Clin. Oncol., 2018, 36(23), 2386- 2394. http://dx.doi.org/10.1200/JCO.2018.77.7672 PMID: 29906251

Gupta SK, Mladek AC, Carlson BL, et al: Discordant in vitro and in vivo chemopotentiating effects of the PARP inhibitor veliparib in temozolomide sensitive versus -resistant glioblastoma multiforme xenografts. Clin Cancer Res 20:3730-3741, 2014

Plummer R, Jones C, Middleton M, Wilson R, Evans J, Olsen A, et al. Phase I study of the poly(ADP-ribose) polymerase inhibitor, AG014699, in combination with temozolomide in patients with advanced solid tumors. Clin Cancer Res (2008) 14:7917–23. doi:10.1158/1078-0432.CCR-08-1223

Daniel RA, Rozanska AL, Mulligan EA, et  al. Central nervous system penetration and enhancement of temozolomide activity in childhood medulloblastoma models by poly(ADP-ribose) polymerase inhibitor AG-014699. Br J Cancer. 2010;103(10):1588–1596.

Round 3

Reviewer 1 Report

The authors did more literature review and further clarified the study design and rationale. I do not have further concern.